# GLLP: Graph Learning from Label Proportions

## Abstract

Learning from Label Proportion (LLP) is a weakly supervised learning paradigm in which only aggregated label proportions over collections of instances (*i.e.,* bags) are provided, rather than individual labels. This allows classification while preserving privacy or reducing annotation costs. Existing LLP methods, however, have been largely restricted to *i.i.d.* tabular or image data. No solution currently addresses graphs, where instances are inherently interdependent through network structure. In this paper, we generalize LLP to the graph domain and study the problem of node classification with label proportions, where only distributional supervision is available for node bags, and the goal is to infer labels for all nodes in the graph. We argue that the lack of node-level supervision is the main challenge for LLP on graphs, and that existing methods based on *i.i.d.* assumptions fail to exploit topological correlations. To overcome this, we propose GLLP (Graph Learning from Label Proportions), a framework that leverages Optimal Transport (OT) with a homophily-aware cost to generate soft pseudo-labels for individual nodes. These pseudo-labels provide stronger supervision signals for training Graph Neural Networks. We further establish theoretical guarantees showing the alignment of our cost function with the node classification objective. Extensive experiments on six homophilic graph benchmarks demonstrate that GLLP consistently outperforms existing LLP baselines and variants. Code and benchmark datasets will be released for public access.

## 1 Introduction

Traditional graph learning problems typically assume access to node-level supervision. For example, in node classification, a subset of nodes in the graph are explicitly labeled, and the goal is to infer the labels of remaining nodes. Graph Neural Networks (GNNs) (Kipf & Welling, 2017) have emerged as powerful models in this setting, leveraging node features and graph structure to propagate supervision across the network. Despite its successes in many applications, such as disease networks (Jha et al., 2022), transportation (Rahmani et al., 2023), and social networks (Awasthi et al., 2023), this paradigm fundamentally relies on explicit node-level labels.

Unfortunately, exposing such fine-grained labels in practice is either infeasible (due to high labeling cost) or undesirable (due to privacy concerns). To wit, in online advertising, it is often prohibitive to disclose personal information of *individual* users; in contrast, ad conversion reporting systems provided by Apple, Google, and Android allow third-party services to access only aggregated conversion statistics across *multiple users* (Busa-Fekete et al., 2023).

This motivates our study of *Graph Learning from Label Proportions* (GLLP), where no individual node label can be observed; instead, the learner observes only collections of unlabeled node feature vectors (called *bags*), together with the proportion of positive examples within each bag. Figure 1 (a) and (b) demonstrate the difference between traditional semi-supervised node classification and our GLLP paradigm, where in our case only bag-level label proportions are available.

One may explore two intuitive methods to solve the GLLP problem. The first assigns pseudo-labels to individual nodes in each bag by sampling according to the ground-truth label proportions. The model is then trained using cross-entropy loss on these pseudo-labels, a method we denote as LLP-PCE. The second employs a Kullback–Leibler (KL) divergence loss to directly align the aggregated GNN predictions with the observed bag-level proportions Ardehaly & Culotta (2017), which we

denote as LLP-KL. Figure 1(c) presents preliminary results comparing these two strategies against a traditional semi-supervised node classifier. Alas, the semi-supervised baseline substantially outperforms both LLP-PCE and LLP-KL, and notably, increasing the number of bags (and the availability of label proportions) does not improve their performance.

We hypothesize from this observation that the main challenge in GLLP lies in the lack of a mechanism to translate coarse bag-level label proportions into reliable node-level supervision *compatible with graph structure*. Simply matching aggregate distributions fails because it ignores how individual nodes, connected through homophily and higher-order structural dependencies, contribute to such aggregate distributions. This necessitates a principled approach to generate surrogate node-level labels that are consistent with bag-level distributions while also respecting graph topology. In response, we propose to decompose GLLP into two subproblems: (i) Optimal Transport (OT) assignment, namely, given a bag-level label distribution and the node-level predictions from GNN models, we solve an OT problem to derive *soft pseudo-labels* for each node. The cost matrix in OT is constructed using graph homophily, ensuring that pseudo-label assignments respect the structural similarity among neighboring nodes. (ii) Node classification, where GNNs are trained on these pseudo-labels as surrogate supervision. Alternating between the OT-based pseudo-label generation and GNN training enables to iteratively refine node-level predictions, while maintaining global consistency with bag-level proportions.

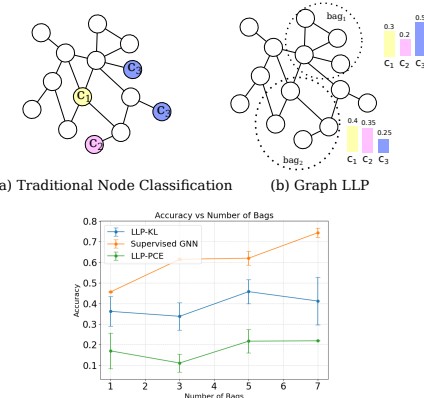

(a) Traditional Node Classification  (b) Graph LLP

(c) Preliminary experiments over weak supervision

Figure 1: Comparison between (a) traditional node classification and (b) graph learning from label proportions (GLLP). In (c), preliminary experiments show that GNNs trained with bag-level KL loss (LLP-KL) or pseudo-label cross-entropy (LLP-PCE) perform significantly worse than supervised GNNs with node-level labels, highlighting the weakness of bag-level supervision alone.

**Specific contributions** of this paper are summarized as follows.

- This is the first study that investigates and formalizes the problem of *graph learning from label proportions* (GLLP), where only bag-level label proportions are available for supervision. We also demonstrate that naïve proportion-matching strategies cannot solve GLLP.
- We propose a novel GLLP approach by solving OT with a homophily-aware cost matrix, where soft pseudo-labels are generated as structurally consistent surrogate supervision at node level. A training scheme that alternates between pseudo-label assignment and GNN updates is devised, improving node-level prediction while preserving bag-level consistency.
- We provide a theoretical analysis showing that the proposed cost function aligns with the node classification objective and derive its performance bound on homophilic graphs.
- Extensive experiments on 6 benchmark graph datasets are carried out, where our proposed GLLP approach outperforms existing LLP competitors by 17% on average in node classification accuracy. Ablation studies confirm the necessity of each proposed component.

## 1.1 RELATED WORKS

**Graph Neural Network (GNN)** has been developed under the message-passing scheme, where each node iteratively aggregates messages from its neighbors to update its own representation. Since Kipf & Welling (2017) proposed the spectral-based Graph Convolutional Network (GCN), which propagates node features through the augmented graph Laplacian, numerous variants have been proposed, including Graph Attention Network (GAT) (Veličković et al., 2018), GraphSAGE (Hamilton et al., 2017), and Graph Isomorphism Network (GIN) (Xu et al., 2019), each distinguished by different aggregation and update mechanisms. For homophilic graphs, Abu-El-Haija et al. (2019) validated that even a shallow GCN can obtain strong classification performance. In this work, we adopt a shallow GCN as the backbone due to its simplicity and effectiveness in homophilic graphs.

**Learning From Label Proportion (LLP)** dates back to classical machine learning like SVM and logistic regression on *i.i.d.* tabular data (Kück & de Freitas, 2005; Musicant et al., 2007; Quadrianto et al., 2009). Early studies focused mainly on binary classification with label proportion matching,

and their solutions were often tightly coupled with specific algorithms (Havaldar et al., 2024). More recently, with the advent of deep learning, attention has shifted to multi-class LLP on image data, with a growing emphasis on individual-level classification performance under proportion-only supervision (Busa-Fekete et al., 2023; Dulac-Arnold et al., 2019). Several directions have emerged. For example, Tsai & Lin (2020) proposed LLP-VAT by adding consistency regularization to the cross-entropy loss; Asanomi et al. (2023) introduced MixBag that mixes samples across bags to form new bags with unbiased label proportion estimations; Liu et al. (2025) devised progressive training with knowledge distillation; and LLP-GAN augments real images with synthetic ones, leveraging adversarial losses to provide additional unsupervised signals (Liu et al., 2019).

To the best of our knowledge, no prior work has investigated LLP in the context of graph-structured data. This paper addresses this gap by extending the LLP paradigm to graphs, where structural dependencies among nodes impose unique challenges beyond those in the *i.i.d.* setting.

## 2 PRELIMINARIES

**Problem Definition** Denoting an undirected graph by $G = (A, V, X, Y)$ with $n$ nodes, where $A \in \{0,1\}^{n \times n}$ is the adjacency matrix, $V$ is the set of vertex, $v_i$ is the node $i$, $X \in \mathbb{R}^{n \times f}$ are the node features with feature dimension $f$, and $Y$ are the corresponding labels for all nodes, and there are $c$ unique classes in total. Denote a set $\mathcal{B} = \{B_1, B_2, ..., B_k\}$ as the set of $k$ bags, with each bag $B_i$ containing $b_i$ number of nodes. Those bags are mutually exclusive. For any bag $B_i$, a label proportion $\boldsymbol{q_i}$ is provided, where its $j$-th entry $q_{ij}$ indicates the proportion of nodes in this bag that belong to label class $j$, and $\|\boldsymbol{q_i}\|_1 = \sum_{j=1}^{c} q_{ij} = 1$. For simplicity, we define a label proportion matrix of all bags as $Q = [\boldsymbol{q_1}, \ldots, \boldsymbol{q_k}]^\top \in [0,1]^{c \times k}$. We denote the subgraph adjacency matrix restricted to the nodes in bag $B_i$ as $A^i \in \{0,1\}^{b_i \times b_i}$. Nodes that do not belong to any bag are treated as the test set, with corresponding ground-truth labels denoted by $Y_{\text{test}}$. Namely, $|Y_{\text{test}}| \ll |Y|$ and $|B_1| + \ldots + |B_k| + |Y_{\text{test}}| = n$. The goal of GLLP is to learn a predictive model $f_\Theta(A, X, V, \mathcal{B}, \mathcal{Q})$ that leverages the graph structure, node features, and bag-level label proportions to predict labels $\tilde{Y}_{\text{test}}$ for the test nodes. The objective is to ensure that the predictions $\tilde{Y}_{\text{test}}$ approximate the ground-truth labels $Y_{\text{test}}$ as closely as possible, despite the absence of individual node-level supervision during training.

**Graph Convolution Network (GCN)** A message-passing GNN learns the representations of nodes as $Z^{l+1} = f_\Theta(A, Z^l)$, where $Z^l$ are the node embeddings in layer $l$. For a simple GCN layer, the message passing function $f_\Theta(\cdot)$ is defined in the following form:

$$f_\Theta(A, Z^l) = \sigma(\tilde{A} Z^l \Theta), \ \ \tilde{A} = D^{-\frac{1}{2}}(A + I)D^{-\frac{1}{2}}, \tag{1}$$

which equates to a node-wise update $Z_i^{l+1} = \sigma(\sum_{j \in \mathcal{N}(i)} \frac{1}{\sqrt{d_i d_j}} Z_j^l \Theta)$, where $\mathcal{N}(i)$ and $d_i$ denote the neighbors and degree of node $v_i$, respectively. $\Theta$ are the learnable parameters that are shared between all nodes, and $\sigma(\cdot)$ is the non-linear activation function such as $\text{Relu}(\cdot)$.

As bag-level label proportion in GLLP is provided as distribution, we explore several loss functions used to gauge distribution-wise similarity or distance.

**Kullback–Leibler (KL) and Cross-Entropy Loss Functions** For the commonly used KL loss, we need to aggregate the soft label distributions for nodes within each bag and use KL divergence over the ground truth label proportion $Q$. For a bag $B_i$, assume its ground-truth label distribution is $Q_i \in \mathbb{R}^c$, and the predicted bag label distribution is denoted by prediction $\tilde{Q}_i \in \mathbb{R}^c$. For graph LLP learning with $k$ labeled bags, the KL loss can be calculated as follows:

$$\mathcal{L}_{KL}(Q, \tilde{Q}) = \sum_{i=1}^{k} Q_i \log(\frac{Q_i}{\tilde{Q}_i}). \tag{2}$$

We observe the relationship between $\mathcal{L}_{KL}$ and the cross entropy loss:

$$\mathcal{L}_{CE}(Q, \tilde{Q}) = -\sum_{i=1}^{k} Q_i \log \tilde{Q}_i = \mathcal{L}_{KL} - \sum_{i=1}^{k} Q_i \log Q_i. \tag{3}$$

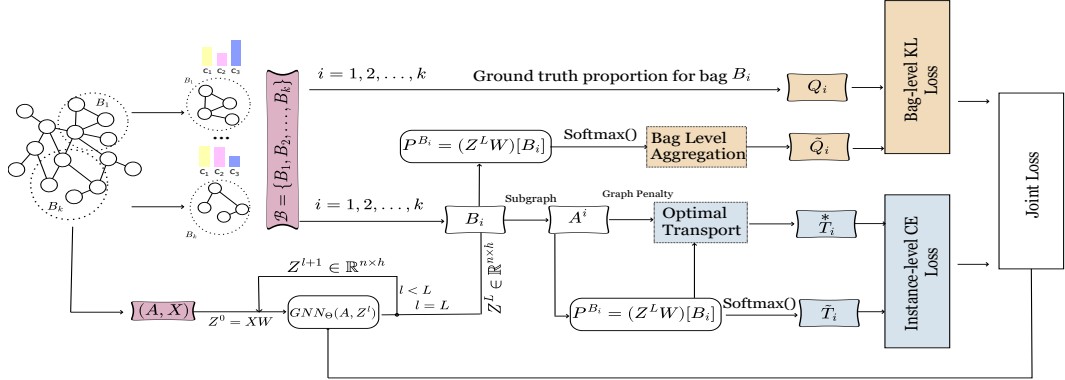

Figure 2: Overview of the proposed GLLP framework. The model integrates two complementary supervision signals: (i) a bag-level KL loss (top orange branch), which enforces consistency between predicted and ground-truth label proportions for each bag, and (ii) an instance-level cross-entropy (CE) loss (bottom blue branch), which leverages soft pseudo-labels obtained via optimal transport. Purple components denote the inputs, including the graph structure, node features, and bag-level label proportions. The overall joint loss in Eq. 11 combines both losses, where the KL loss is defined in Eq. 2 and the instance-level CE loss in Eq. 10.

**Optimal Transport Loss** Instead of measuring information difference between two distributions, like KL-divergence does, Optimal Transport (OT) intends to find the minimum costs to transform one probability distribution into another. We formulate the OT subproblem over a single bag $B_i$. Formally, define two distributions with $\mathbf{a} \in \mathbb{R}^{b_i}$ the distribution over the nodes in the bag and $\mathbf{b} \in \mathbb{R}^c$ the bag's class distribution. Define a transport plan $T \in \mathbb{R}_+^{b_i \times c}$ with the constraints that $T\mathbf{1}_{b_i} = \mathbf{a}$, $T^\intercal \mathbf{1}_c = \mathbf{b}$, denote $U(\mathbf{a}, \mathbf{b})$ the set that has matrix satisfies such conditions as the set of admissible couplings for $\mathbf{a}$ and $\mathbf{b}$. A cost matrix $C \in \mathbb{R}^{b_i \times c}$ is equipped with the goal of OT for the transport plan $T$ to become a minimization problem:

$$\mathcal{L}_C(\mathbf{a}, \mathbf{b}) = \min_{T \in U(\mathbf{a}, \mathbf{b})} \langle T, C \rangle = \min_{T \in U(\mathbf{a}, \mathbf{b})} \sum_{i,j} T_{ij} C_{ij}, \tag{4}$$

which defines the original Kantorovich's OT problem (Monge, 1781). Following the previous work Liu et al. (2025); Cuturi (2013), we leverage its entropic variant, defined as

$$\mathcal{L}_C^\epsilon(\mathbf{a}, \mathbf{b}) = \min_{T \in U(\mathbf{a}, \mathbf{b})} \langle T, C \rangle - \epsilon H(T), \quad \text{where} \quad H(T) = -\sum_{i,j} T_{i,j} \log(T_{i,j})). \tag{5}$$

We initiate $\mathbf{a}$ as an unbiased uniform distribution. Note that the key of the OT problem is the cost function design. In general, we would like to connect the cost function back to the final objective of the node classification task and details are presented in our proposed method in the next section.

## 3 GLLP: GRAPH LEARNING FROM LABEL PROPORTIONS

The framework of the proposed GLLP algorithm is shown in Figure 2. The pseudo code of the proposed algorithm are reported in Algorithm 1 in Appendix. The limitation of direct KL loss for GLLP motivates our proposed design. The key idea is to employ an alternating training scheme that extracts stronger instance-level supervision from bag-level proportions by leveraging graph topology through optimal transport (OT). Specifically, the OT plan maps bag-level label distributions to soft pseudo-labels for individual nodes, producing richer supervisory signals than raw proportion matching. These pseudo-labels are then used in a soft cross-entropy (CE) loss to guide the training of a GCN model. In turn, the GCN generates node-level logits for each bag, which are fed back into the OT module, creating a closed loop between pseudo-label assignment and node classification.

### 3.1 NODE-LEVEL SUPERVISION SIGNAL DERIVATION

In order to minimize the label ambiguity and derive strong supervision signal, we propose to use optimal transport equipped with a cost matrix consists of a graph penalty cost and a negative log probability cost based on current model output and subgraph topology within a bag.

**Bag Label Distribution Prediction** In general, an $L$ layer GCN is leveraged as a backbone model. After the final GCN layer, a linear decoder layer projects the embedding $Z^L$ to the output dimension c as raw logits for all nodes $P = Z^L W$ with $W$ the projectable projection parameters that project the embedding dimension to the output dimension c. Note that since our learning setting is transductive, all the node features including test nodes will be involved in the information exchange stage Eq. 1 but the final loss guided will be training nodes (in our case, nodes in the bag) only.

For one bag $B_i$, when there is no confusion, we will use the same symbol $B_i$ as its nodes indices within the bag for simplicity and we can define a logit at the bag-level as $P^{B_i} = P[B_i]$ where $P^{B_i} \in \mathbb{R}^{b_i, c}$. For a bag $B_i$, We use the simple mean aggregation to obtain bag-level prediction $\tilde{Q}_i = \frac{1}{b_i} \sum_{j=1}^{b_i} softmax(P_j^{B_i})$ where $P_j^{B_i}$ is the soft label distribution for node $j$ in the bag $B_i$ and $\tilde{Q}_i$ is the predicted bag label proportion.

**Cost Matrix of entropic OT for soft pseudo-label generation** To fully exploit the graph topology and utilize the graph smoothness assumption in homophily graph to produce a more accurate node-level pseudo-labels from ground-truth bag-level label proportion. A linear graph neighbor averaging cost is added in addition to the negative log probability term for the basic cost function used related to the model output. Specifically, for one bag $B_i$ with node logit output as $P^{B_i}$ within the bag, we first compute its normalized probability $\tau$ as:

$$Q^\tau = softmax_\tau(P^{B_i}), \tag{6}$$

where $Q^\tau$ is the normalized probability for each node in the bag with temperature $\tau$ controlled the smoothness of the distribution. Define the base cost function $C^b(Q^\tau)$ as:

$$C^b(Q^\tau) = \sum_{i=1}^{b_i} -\log(Q^\tau), \tag{7}$$

where $C^b$ leverage the negative log probability of nodes in the bag as the base cost measure.Next, the graph penalty cost function $C^g$ takes the node's probability and its topology relation within bags as the graph measure for the cost matrix:

$$C^g(Q^\tau, A^i) = D^{-1} A^i Q^\tau, \tag{8}$$

where $A^i$ is the subgraph induced from the nodes within the bag $B_i$ and $D^{-1} A^i$ is its normalized form in rows. Finally, we obtain the cost matrix $C$ through the cost function $C^\lambda$ and control the graph penalty cost through the coefficient $\lambda$ as a hyperparameter, defined as

$$C = C^\lambda(Q^\tau, A^i) = C^b(Q^\tau) - \lambda C^g(Q^\tau, A^i). \tag{9}$$

After obtaining the cost matrix $C$, we can plug it into the minimization problem as defined in Eq. 5. Solving the problem is equivalent to searching for an optimal transport plan $T$ that satisfies all the constraints and minimizes its distance from the cost matrix $C$. In Fig 3, we show how to compute the cost matrix $C$ given the original graph data and an example bag and outputs the optimal transport $\overset{*}{T}$. We show the constraints that the transport matrix $T$ satisfies with its row and column margin equal to $\mathbf{1}$ and the bag distribution $\mathbf{q}$.

## 3.2 Bag-Level Loss Assessment

Equipped with the pseudo-label output by OT process, we can compute instance-level supervision loss for bag $B_i$ as:

$$\tilde{T}_i = softmax(P^{B_i}), \ \mathcal{L}_{CE}(\overset{*}{T}, \tilde{T}) = -\sum_{j=1}^{b_i} \overset{*}{T}_j \log \tilde{T}_j, \tag{10}$$

where $\tilde{T} = Q^{\tau=1}$ is the predicted instance-level supervision and $\overset{*}{T}$ is the pseudo-label we obtained from OT. In practice, we row normalized $\overset{*}{T}$ with the uniform distribution over number of nodes per bag to ensure the pseudo-label is row-stochastic.

Figure 3: Optimal Transport outputs soft pseudo label that minimizes in terms of a cost matrix $C$ and satisfies marginal condition. Step ① applies the $\tau$ softmax normalization for the bag $B_i$ over the predicted logits from GNN model to obtain $Q^\tau \in \mathbb{R}^{b_i, c}$. Step ② then leverages $Q^\tau$ along with the intra-bag topology $A^1 \in \mathbb{R}^{b_i, b_i}$ to compute the cost matrix $C \in \mathbb{R}^{b_i, c}$ with the Eq. 9. Step ③ performs the OT algorithm to solve the constrained entropic minization problem following Eq. 5 and obtain the optimal solution. Finally, Step ④ outputs the soft pseudo-label by another softmax normalization to ensure that it is a valid distribution.

**Loss Functions of GCN model for node classification task**    Two types of losses are jointly optimized over the GCN model with the joint loss defined as follows:

$$\mathcal{L}_{ours}(Q, \tilde{Q}, \overset{*}{T}, \tilde{T}) = (1 - \beta)\mathcal{L}_{KL}(Q, \tilde{Q}) + \beta\mathcal{L}_{CE}(\overset{*}{T}, \tilde{T}), \tag{11}$$

where $\mathcal{L}_{KL}$ follows Eq. 2 and $\mathcal{L}_{CE}$ follows Eq. 10. We obtain a joint loss over the bag-level supervision and the node-level supervision with $\overset{*}{T}$ computed from OT. While $\overset{*}{T}$ provide strong node-level supervision, it might distort the ground truth bag-level distribution if not sufficiently converged. To maintain an efficient OT problem solver and also respect the ground-truth distribution signal, we therefore use a $\beta$ coefficient to jointly regularize each other to achieve a more robust result. We treat the $\beta$ as a hyperparameter and provide an ablation study to show its effect in the empirical study.

### 3.3 THEORETICAL ANALYSIS

To incorporate graph topology into the OT learning and obtain stronger node-level pseudo labels, we introduce a linear graph penalty term. To justify the merit of this graph-regularized cost function over the vanilla base cost (negative log-probability), we establish Theorem 1, which guarantees that under homophilic graphs, the graph-penalized cost yields pseudo-labels that are closer in expectation to the ground-truth labels than those obtained using the base cost alone.

**Theorem 1.** *Consider one bag of $n$ nodes with subgraph adjacency $A$. Let $R = D^{-1}A$ denote the row normalized adjacency matrix, $c$ the number of unique classes, and $\mathbf{q}$ the normalized label proportion of the bag. The graph-regularized entropic OT problem can be defined as:*

$$\min_T \langle C^\lambda, T \rangle + \epsilon \sum_{i=1}^{b_i} \sum_{j=1}^{c} T_{ij} log(T_{ij}), \tag{12}$$

*subject to $\sum_{j=1}^{c} T_{ij} = 1$, $\frac{1}{n}\sum_{i=1}^{n} T_i = q$. With Lagrangian multipliers $\alpha$ and $\kappa$, the solution is:*

$$\epsilon \log(T_{ij})(\lambda) = \alpha_i + \kappa_j - C_{ij}^b + \lambda(RQ^\tau)_{ij}, \tag{13}$$

*which, in compact matrix form, becomes*

$$\overset{*}{T}(\lambda) = Diag(\alpha) \exp\left(-\frac{C^b}{\epsilon}\right) \exp\left(\frac{\lambda}{\epsilon}RQ^\tau\right) Diag(\kappa), \tag{14}$$

*where $Diag(\cdot)$ denotes a diagonal matrix. Given the true label $Q_i^*$ for node $i$ in the bag, $r = softmax(\gamma RQ^\tau)$ for some $\gamma$, and $\tilde{Y}_i(\lambda = 0)$ the normalized pseudo label of $\overset{*}{T}(\lambda = 0)$. Under the homophily assumption, we have*

$$\mathbb{E}[KL(Q_i^*||r)] \leq \mathbb{E}[KL(Q_i^*||\tilde{Y}_i(\lambda = 0))]. \tag{15}$$

*Consequently, for $\tilde{Y}_i(\lambda)$ denoting the normalized pseudo-label with $\lambda > 0$, we obtain*

$$\mathbb{E}[KL(Q_i^*||\tilde{Y}_i(\lambda > 0))] \leq \mathbb{E}[KL(Q_i^*||\tilde{Y}_i(\lambda = 0))]. \tag{16}$$

**Remark** The theoretical benefit arises from the interaction between graph topology and the entropic OT solution, as indicated by the structure of Eq. 14. The linear graph penalty $\lambda(RQ^\tau)$ in the cost matrix $C^\lambda$ induces exponential reweighting in the transport plan $\overset{*}{T}(\lambda)$. When the pseudo-label $\tilde{Y}_i(\lambda)$ is obtained from $\overset{*}{T}(\lambda)$, the logarithmic form of the KL divergence converts this reweighting into a convex inequality. This bounds the result in Eq. 16, which establishes that, under homophilic assumption, $\tilde{Y}_i(\lambda > 0)$ provides a strictly better approximation to the true label distribution $Q_i^*$ than $\tilde{Y}_i(\lambda = 0)$ obtained from the vanilla base cost $C^b$. As a result, the alternating training scheme (Figure 2) can exploit these higher-quality pseudo-labels to achieve strong node classification performance, even in the absence of individual-level labels. Proof of Theorem 1 is provided in the Appendix. Our empirical results across 6 benchmark graphs corroborate these theoretical findings.

# 4 EXPERIMENTS

To verify the effectiveness of our proposed method, we conduct empirical studies over 6 homophilic graph datasets including Cora, CiteSeer, PubMed from Planetoid Dataset Yang et al. (2016), Amazon-Photos, Amazon-Computers from Amazon Dataset Shchur et al. (2019) (we denote them as Photos and Computers in the latter work for simplicity), and WikiCS Datset Mernyei & Cangea (2020). All datasets are originally node classification task and in the graph LLP setting, we sample the bag of nodes with random iid sampling strategy and produce data splits with different number of bags in the range of $[3, 5, 7, 10]$ and different number of nodes within bags ranging from $[10, 20, 30, 40, 50, 60, 70, 80]$. All experiments are conducted over 5 random seeds and we report its 5-times average results with standard deviation.

**LLP Baselines.** As this is the first study working on LLP learning on graph, we select several recent baselines from existing LLP settings on image and tabular dataset to compare with our proposed methods, including LLP-MixBag Asanomi et al. (2023), PT-LLP Liu et al. (2025),LLP-VAT Tsai & Lin (2020), and LLP-KL Ardehaly & Culotta (2017) which directly leverages KL loss as supervision signals. For all baseline models, we use the simple 2-layer graph convolution network training model as it has been shown to be effective in terms of homophilic graphs.

- LLP-KL: For each bag, we aggregate the predictions for nodes within the bag and use KL loss over the ground-truth distribution to guide the GCN training.

- LLP-VAT: For each node in the bag, following Tsai & Lin (2020), we generate the noisy embeddings thorough Virtual Adversarial Training (VAT) Miyato et al. (2019) and compare it with the original embeddings with a distance measure, which computes the consistency loss, together with KL loss, the total consistency regularized KL loss acts as the loss function for GCN training.

- LLP-MixBag: For two randomly sampled bag, we sample nodes within each bag to form a new augmented bags and compute the proportion according to Asanomi et al. (2023), finally, the original bag KL loss together with the augmented bag unbiased KL loss acts as supervision for GCN training.

- PT-LLP: following Liu et al. (2025), a GCN with KL loss is trained to output its soft pseudo label. Then, the soft pseudo label is transferred as the starting point for OT training and a new GCN model is trained using the OT's output pseudo label with CE loss. The knowledge distillation in the middle stage is ignored as we observe that the knowledge distillation process significantly degrades the model performance, leading to poor pseudo-label quality.

As an ablation test to show that the graph signal should be inserted in the OT stage but not directly jointly optimized with KL loss, we showed an LLP-KLG variant which explicitly adds a graph penalty cost jointly optimized with the bag-level KL loss to train the GCN model.

**Datasets & Experiment Setup** Table 1 shows the statistics of each dataset. Our setting is multi-class node classification and we have a variety of different class number ranging from 3 classes to 10 classes. For hyperparameter searching, we have $\lambda, \beta, \tau$ as the primary hyperparameters. We fixed the temperature $\tau = 2$ thorough the experiments and search $\lambda$ in the range $[0.5, 1]$ and $\beta$ in

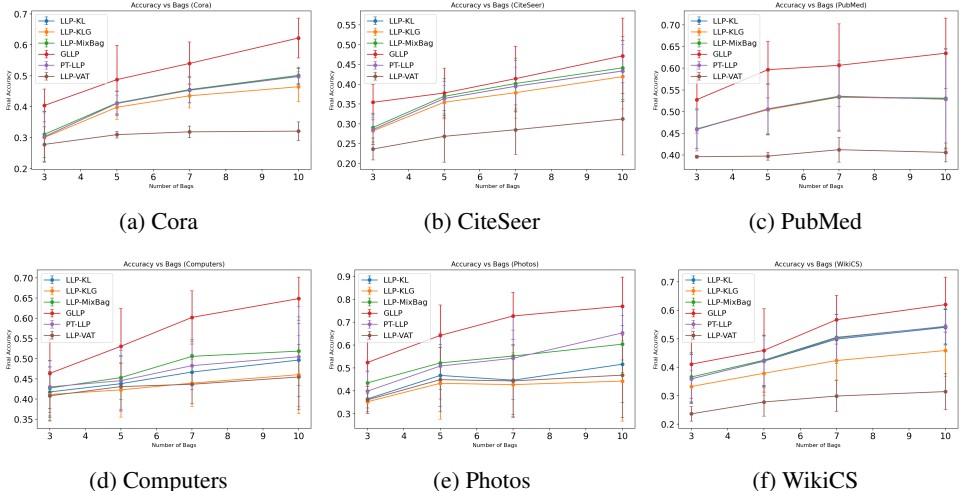

(a) Cora      (b) CiteSeer      (c) PubMed

(d) Computers      (e) Photos      (f) WikiCS

Figure 4: Test Accuracy versus number of bags for six datasets. The $x$-axis shows the number of bags labeled and $y$-axis shows the average final test accuracy over five seeds. Each method is plotted with mean and standard deviation bar.

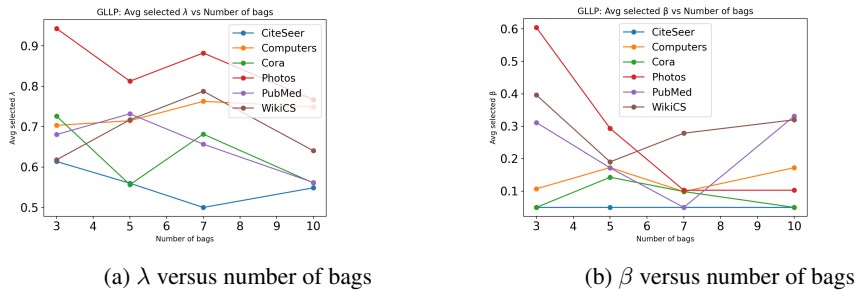

(a) $\lambda$ versus number of bags      (b) $\beta$ versus number of bags

Figure 5: Best hyperparameter value OT coefficient $\beta$ and graph penalty cost coefficient $\lambda$ on average over different number of bags for GLLP.

the range $[0.05, 0.5, 1]$. Note that due to the lack of validation dataset, we choose the combination that achieves the highest accuracy. It is noticed that the accuracy is sufficiently stable for a specific groups of combinations and other combinations show clearly low acc results. Therefore, in practice, only a few sample labels could suffice to select the proper hyperparameter. In our experiments, we early stop the model training if the training loss doesn't decrease for a fixed number of epochs.

**Results** Table 2 shows the averaged test accuracy for each baseline methods and our proposed methods over 5 random seeds and across various number of nodes per bag and various number of bags. For all datasets, GLLP shows significant performance boost compared with the second best results, verifying the effectiveness of our proposed method and our theoretical analysis.

Except our method, we notice that LLP-MixBag consistently shows some improvements compared with other methods, suggesting its general effectiveness. However, due to the lack of graph signals, the improvements is marginal compared with GLLP.

Table 1: Dataset Statistics with number of nodes, edges, features and unique classes.

|  | Cora | CiteSeer | PubMed | Computers | Photos | WikiCS |
|---|---|---|---|---|---|---|
| # of Nodes | 2,708 | 3,327 | 19,717 | 13,752 | 7,650 | 11,701 |
| # of Edges | 5,429 | 4,732 | 44,338 | 491,722 | 238,162 | 216,123 |
| # of Features | 1,433 | 3,703 | 500 | 767 | 745 | 300 |
| # of Classes | 7 | 6 | 3 | 10 | 8 | 10 |

Table 2: Our proposed method compared with different baseline methods over six datasets averaged across all combinations of bag size and nodes per bag.

| Methods | Cora | CiteSeer | PubMed | Computers | Photos | WikiCS |
|---|---|---|---|---|---|---|
| LLP-KL | $0.416 \pm 0.088$ | $0.369 \pm 0.078$ | $0.507 \pm 0.080$ | $0.455 \pm 0.071$ | $0.449 \pm 0.138$ | $0.455 \pm 0.105$ |
| LLP-VAT | $0.307 \pm 0.037$ | $0.276 \pm 0.068$ | $0.403 \pm 0.019$ | $0.433 \pm 0.062$ | $0.430 \pm 0.143$ | $0.460 \pm 0.103$ |
| LLP-MixBag | $0.420 \pm 0.086$ | $0.376 \pm 0.079$ | $0.508 \pm 0.080$ | $0.476 \pm 0.072$ | $0.528 \pm 0.118$ | $0.457 \pm 0.106$ |
| PT-LLP | $0.416 \pm 0.088$ | $0.369 \pm 0.078$ | $0.507 \pm 0.083$ | $0.466 \pm 0.086$ | $0.526 \pm 0.160$ | $0.398 \pm 0.085$ |
| LLP-KLG | $0.400 \pm 0.081$ | $0.359 \pm 0.075$ | $0.507 \pm 0.080$ | $0.434 \pm 0.072$ | $0.414 \pm 0.137$ | $0.282 \pm 0.056$ |
| GLLP | $\mathbf{0.513} \pm \mathbf{0.109}$ | $\mathbf{0.405} \pm \mathbf{0.083}$ | $\mathbf{0.591} \pm \mathbf{0.086}$ | $\mathbf{0.561} \pm \mathbf{0.102}$ | $\mathbf{0.666} \pm \mathbf{0.147}$ | $\mathbf{0.514} \pm \mathbf{0.138}$ |

For PT-LLP, although it has a similar OT component as we proposed, we observe that its performance gain is unstable thorough six datasets with several performance maintained to be the same as direct LLP-KL training and with Computers and Photos improved performance but degraded in WikiCS. Compared with alternative training scheme as proposed in GLLP, we argue that PT-LLP is not flexible and stable across various datasets.

For LLP-VAT, we observe that its performance degrades on five out of six datasets and we hypothesize that this could be due to the lack of graph information when generating adversarial examples as VAT only consider node features as input.

As an ablation comparison of whether direct graph regularization cost along with KL loss can be useful for GCN model training, we observe that LLP-KLG shows no improvements or even degrades significantly on several datasets, suggesting that direct graph regularization is not sufficient for the Graph Label Proportion Learning setting and the task is non-trivial to investigate.

**Ablation Study**    To compare the effect of the number of bags and node size per bags on GLLP and baseline methods, we show in Figure 4 that our methods consistently outperforms the other baselines across all datasets, suggesting the robustness of our method in terms of different resources settings. For all datasets in terms of the increase of the number of bags, GLLP shows a consistent performance increase similar to learning from direct node label signal while other methods show inconsistent performance despite the supervision resources increase. This implies that our method can convert bag signal to stronger node-level supervision and utilize the signal effectively and scalably.

In Figure 5, we collect best hyperparameter configuration for GLLP with its main hyperparameters $\lambda$ and $\beta$ and compute the average results of their values in terms of number of bags. Specifically, for number of bags, we compute the average results across all numbers of nodes per bag and all seeds. It can be observed that both hyperparameters show more consistent patterns in terms of the number of bags. As the number of bags directly identify the amount of information provided directly, the consistent patterns suggests that our hyperparameters and our proposed supervision is correctly leveraging the resources as much as possible.

## 5    CONCLUSIONS AND LIMITATIONS

This study explores a new learning problem by generalizing label proportion learning onto graphs. Our preliminary studies reveal that directly relying on weak bag-level supervision is insufficient, and that converting such coarse signals into stronger node-level supervision is essential for effective GLLP. To address this, we propose an alternating training scheme that decomposes the task into two coupled subproblems: pseudo-label generation and node classification. Guided by theoretical analysis, we design a homophily-aware cost function that incorporates graph topology into the optimal transport process, yielding high-quality soft pseudo-labels that substantially enhance model performance. Extensive experiments on six benchmark graphs with varying numbers of classes and supervision conditions validate the scalability and effectiveness of the proposed approach.

While our framework demonstrates strong performance, it mainly relies on the homophilic graph assumption and has not been extended to heterophilic graphs with more complex relational patterns. Future work will explore designing cost matrices tailored to diverse graph topologies and integrating GNN architectures specifically developed for heterophilic settings, which can broaden the applicability of GLLP to real-world networks with richer graph structural dynamics.

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

## A APPENDIX

### A.1 THEOREM PROOF

**Theorem.** *For one bag of $n$ nodes, with a subgraph $A$ built upon $n$ nodes in the bag. Denote $R = D^{-1}A$ as row normalized adjacency matrix. $c$ is the number of unique class. Denote $q$ as the normalized label proportion for the bag. A graph regularized entropic Optimal Transport problem can be defined as:*

$$\min_T \langle C^\lambda, T \rangle + \epsilon \sum_{i=1}^{b_i} \sum_{j=1}^{c} T_{ij} log(T_{ij}) \tag{17}$$

*subject to $\sum_{j=1}^{c} T_{ij} = 1$, $\frac{1}{n}\sum_{i=1}^{n} T_i = q$. With $\alpha$, $\kappa$, the Lagrangian multiplier, the solution for Eq. 12 is:*

$$\epsilon \log(T_{ij})(\lambda) = \alpha_i + \kappa_j - C_{ij}^b + \lambda(RQ^\tau)_{ij} \tag{18}$$

*rearranging with matrix compact form, the solution $\overset{*}{T}$ can be shown as:*

$$\overset{*}{T}(\lambda) = Diag(\alpha)\exp\left(-\frac{C^b}{\epsilon}\right)\exp\left(\frac{\lambda}{\epsilon}RQ^\tau\right)Diag(\kappa) \tag{19}$$

*where $Diag(\cdot)$ is the diagonal matrix with $\cdot$ as the diagonal values. Given the true label $Q_i^*$ for node $i$ in the bag, denote $r = softmax(\gamma RQ_i^\tau)$ for some $\gamma$, $Q^\tau$ and $\tilde{Y}_i(\lambda = 0)$ the normalized pseudo label of $\overset{*}{T}(\lambda = 0)$. Under the Homophily graph, we assume the following:*

$$\mathbb{E}[KL(Q_i^*||r)] \leq \mathbb{E}[KL(Q_i^*||\tilde{Y}_i(\lambda = 0))] \tag{20}$$

**Remark**   *We justify the assumption as following: if the graph is homophilic, neighbors of the node $v_i$ tend to share same labels as $v_i$, As a result, aggregation over the average of the neighbor soft label distribution could better represent its true distribution information compared with its own raw prediction.*

*Finally, denote $\tilde{Y}(\lambda)$ as the normalized soft pseudo-label, we can conclude following the assumption in Eq. 15 as:*

$$\mathbb{E}[KL(Q_i^*||\tilde{Y}_i(\lambda > 0))] \leq \mathbb{E}[KL(Q_i^*||\tilde{Y}_i(\lambda = 0))] \tag{21}$$

*Proof.* We first show that the solution of the minimization problem as defined in Eq. 12 has the form shown in Eq. 14.

Denote $\alpha$, $\kappa$ the Lagrangian multipliers for the constraint $\sum_{j=1}^{c} T_{ij} = 1$ and $\frac{1}{n}\sum_{i=1}^{n} T_i = q$. We can obtain the equation as:

$$\frac{\partial(\langle(C^\lambda, T\rangle + \epsilon\sum_{i=1}^{b_i}\sum_{j=1}^{c} T_{ij}log(T_{ij}) + \alpha(1 - \sum_{j=1}^{c} T_{:j}) + \kappa(q - \frac{1}{n}\sum_{i=1}^{n} T_{i:}))}{\partial T_{ij}} = 0 \tag{22}$$

By substituting $C^\lambda$ with Eq. 9 and Solve the partial derivative over $T_{ij}$, and we can obtain the Eq. 14, which suggests that $\overset{*}{T}(\lambda)$ is exponentially reweighted by the graph linear penalty term with $Diag(\alpha)$ and $Diag(\kappa)$ ensures the solution preserves the constraints. Focus on a single row $\overset{*}{T}_i(\lambda)$, which represents the soft pseudo-label (i.e., class distribution) for node $i$. Drop the diagonal scalings momentarily and isolate the exponential kernel:

$$\overset{*}{T}_i(\lambda) \propto \exp\left(-\frac{C_{i\cdot}^b}{\epsilon}\right) \circ \exp\left(\frac{\lambda}{\epsilon}(RQ^\tau)_{i\cdot}\right)$$

At $\lambda = 0$, the solution becomes:

$$\overset{*}{T}_i(0) \propto \exp\left(-\frac{C_{i\cdot}^b}{\epsilon}\right)$$

Therefore:

$$\overset{*}{T}_i(\lambda) \propto \overset{*}{T}_i(0) \circ \exp\left(\frac{\lambda}{\epsilon}(RQ^\tau)_{i\cdot}\right) \tag{23}$$

This is a multiplicative reweighting of the baseline transport $\overset{*}{T}_i(0)$ by a graph-aware factor. Now, we make a change of variable using t for $\frac{\lambda}{\epsilon}$

$$t := \frac{\lambda}{\epsilon\gamma} \quad \text{with} \quad \gamma \geq \frac{\lambda}{\epsilon} \Rightarrow t \in [0, 1]$$

Now:

$$\frac{\lambda}{\epsilon} = t\gamma \quad \Rightarrow \quad \exp\left(\frac{\lambda}{\epsilon}RQ^\tau\right) = \exp\left(t \cdot \gamma RQ^\tau\right)$$

Hence:

$$\overset{*}{T}_i(\lambda) \propto \overset{*}{T}_i(0) \circ \exp\left(t \cdot \gamma(RQ^\tau)_{i\cdot}\right)$$

To make $\overset{*}{T}_i(\lambda)$ a valid soft label distribution, we need to apply softmax normalization:

$$\tilde{Y}_i(t) := \text{Softmax}\left(\log \overset{*}{T}_i(0) + t \cdot \gamma(RQ^\tau)_{i\cdot}\right)$$

Let the second probability distribution (which captures the graph influence) be:

$$r := \text{Softmax}(\gamma(RQ^\tau)_{i\cdot})$$

We know that $log(r)$ is equivalent to $\gamma(RQ^\tau)_{i\cdot}$ up to a constant offset, which can be canceled when put into the softmax function. To ensure that when t=1, the softmax produce the desired endpoint distribution r, we rescale $\tilde{Y}_i(t)$ to:

$$\tilde{Y}_i(t) := \text{Softmax}\left((1-t)\log \overset{*}{T}_i(0) + t \cdot log(r)\right)$$

Then the final soft label distribution $\tilde{Y}_i(t)$ can be expressed as an exponential convex combination of the baseline transport $\overset{*}{T}_i(0)$ and the graph probability distribution $r$:

$$\tilde{Y}_{ik}(t) = \frac{\overset{*}{T}_{ik}(0)^{1-t} \cdot r_k^t}{\sum_j \overset{*}{T}_{ij}(0)^{1-t} \cdot r_j^t} \tag{24}$$

which gives us the final distribution in terms of t. $KL(\mathbf{a}||\mathbf{b})$ follows the form as:

$$KL(\mathbf{a}||\mathbf{b}) = \mathbf{a}log(\frac{\mathbf{a}}{\mathbf{b}})$$
$$= \mathbf{a}log(\mathbf{a}) - \mathbf{a}log(\mathbf{b}) \tag{25}$$

For $KL(q_i^*||\overset{*}{T}(\lambda > 0)$, we can expand and rearrange the normalized solution following Eq. 25 to obtain:

$$KL(Q_i^*||\tilde{Y}_i(t)) = Q_i^*log(Q_i^*) - Q_i^*log(\tilde{y_i}(t)) \tag{26}$$
$$= Q_i^*log(Q_i^*) - (1-t)Q_i^*log(\overset{*}{T}_i(0)) - tQ_i^*log(r) + logZ(t) \tag{27}$$

where $Z(t)$ is the normalized term. It can be observed in Eq. 27, the first term is a constant independent of t, the second and third term are linear in t and the last term is log sum exponential function of t which is convex in terms of t. Therefore, $KL(Q_i^*||\tilde{Y}_i(t)$ is convex function in terms of $t$. Since we can obtain the relation of $log(\tilde{Y}(t))$ in terms of $\overset{*}{T}_i(0))$ and r(k) as:

$$log(\tilde{Y}(t)) = (1-t)log(\overset{*}{T}_i(0)) + tlog(r) \tag{28}$$

Substitute Eq. 28 into Eq 27 and using convex inequality we can obtain:

$$KL(Q_i^*||\tilde{Y}_i(t)) \leq (1-t)KL(Q_i^*||\tilde{Y}_i(0)) + tKL(Q_i^*||r) \tag{29}$$

Leveraging the assumption in Eq. 15, we can therefore obtain the final conclusion:

$$KL(Q_i^*||\tilde{Y}_i(t)) \leq KL(Q_i^*||\tilde{Y}_i(0)) \tag{30}$$

$\square$

## A.2 DETAILED IMPLEMENTATIONS

We provide the pseudo code in Algorithm 1 to show the full pipeline of GLLP.

---

**Algorithm 1: GLLP**

---

**Input:** An undirected graph $G = (A, V, X), \mathcal{B}, \mathcal{Q}$, hyperparameters: $\lambda$, $\beta$, $\tau$, $Epochs$
**Output:** The suitably trained model $\text{GNN}_\Theta(A, X, V, B, Q)$
Initialize the GNN model $\text{GNN}_\Theta(\cdot)$ ;
Initialize a Encoder layer with matrix $W$ ;
**for** $epoch \leftarrow 0$ **to** $Epochs$ **do**
    $Z^0 \leftarrow XW$ ;
    **for** $l \leftarrow 0$ **to** $L - 1$ **do**
        $Z^{l+1} \leftarrow f_\Theta(A, Z^l)$;
    **for** $i \leftarrow 1$ **to** $k$ **do**
        // Compute bag-level loss as following
        Obtain $P^{B_i}$ by softmax normalization over raw logits;
        Obtain $\tilde{Q}^i$ with $P^{B_i}$ using mean aggregation;
        Compute $\mathcal{L}_{KL}(Q_i, \tilde{Q}_i)$ following Eq. 2 ;
        // Compute node-level loss as following
        Obtain subgraph $A_i$ based on $A$ and nodes index within the bag $B_i$ ;
        Compute $\tilde{T}$ with $P^{B_i}$ with $\tau$ softmax normalized logits ;
        Compute $Q^\tau$ with $P^{B_i}$ following Eq. 6 ;
        Compute base cost $C^b(Q^\tau)$ with $Q^\tau$ following Eq. 7 ;
        Compute graph penalty cost $C^g(Q^\tau)$ with $A_i, Q^\tau$ following Eq. 8;
        Compute the total cost $C^\lambda(Q^\tau)$ with $\lambda, C^g(Q^\tau), C^b(Q^\tau)$ following Eq. 9;
        Apply OT optimization with cost $C^\lambda(Q^\tau)$ following entropic sinkhorn iterative optimization and obtain pseudo label $\overset{*}{T}$ ;
        Compute $\mathcal{L}_{CE}(\overset{*}{T}_i, \tilde{T}_i)$ following Eq. 10 ;
        // Compute joint loss
        Compute $\mathcal{L}_{ours}$ with $\mathcal{L}_{KL}, \mathcal{L}_{CE}$, and $\beta$ ;
    Average loss over all bags and backward propagation;
    $\text{GNN}_\Theta(\cdot) \leftarrow$ Update GNN model's parameters.
**return** $GNN_\Theta(\cdot)$;

---

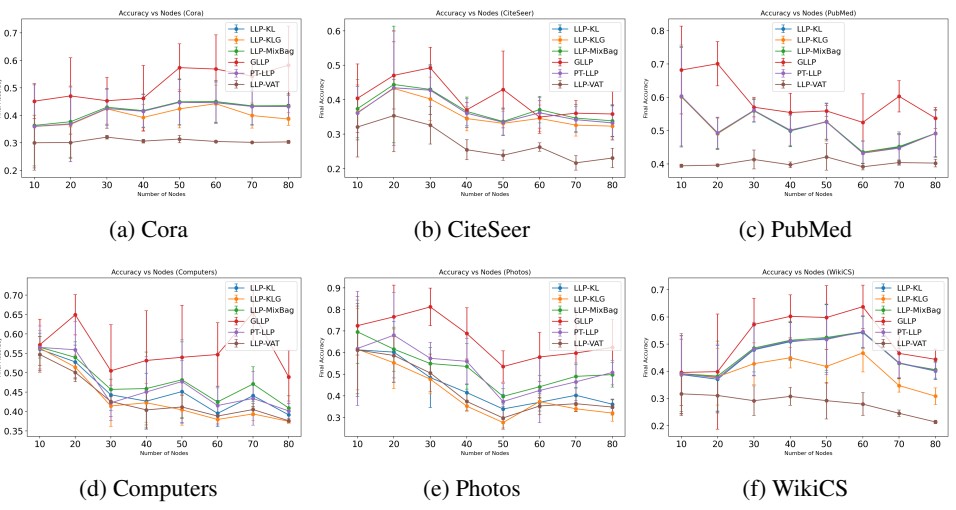

(a) Cora      (b) CiteSeer      (c) PubMed

(d) Computers      (e) Photos      (f) WikiCS

Figure 6: Test Accuracy versus number of nodes per bag for six datasets. The $x$-axis shows the number of nodes per labeled bag and the $y$-axis shows the average final test acc over five seeds. Each method is plotted with mean and standard deviation bar.

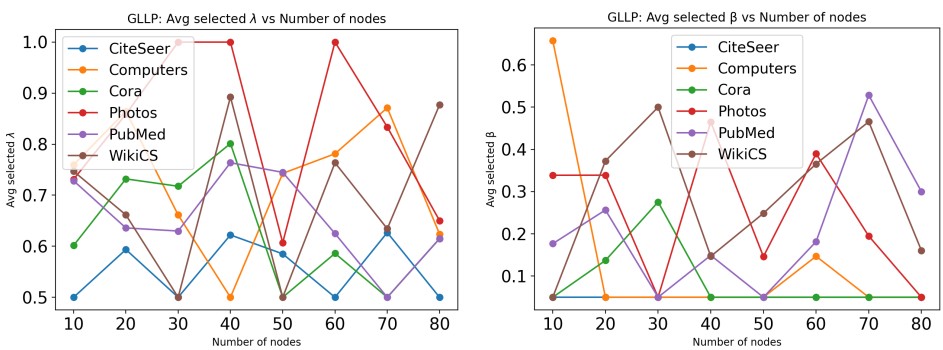

(a) $\lambda$ versus different number of nodes per bag

(b) $\beta$ versus different number of nodes per bag

Figure 7: Ablation results over different number of Nodes in terms of $\lambda$ and $\beta$.

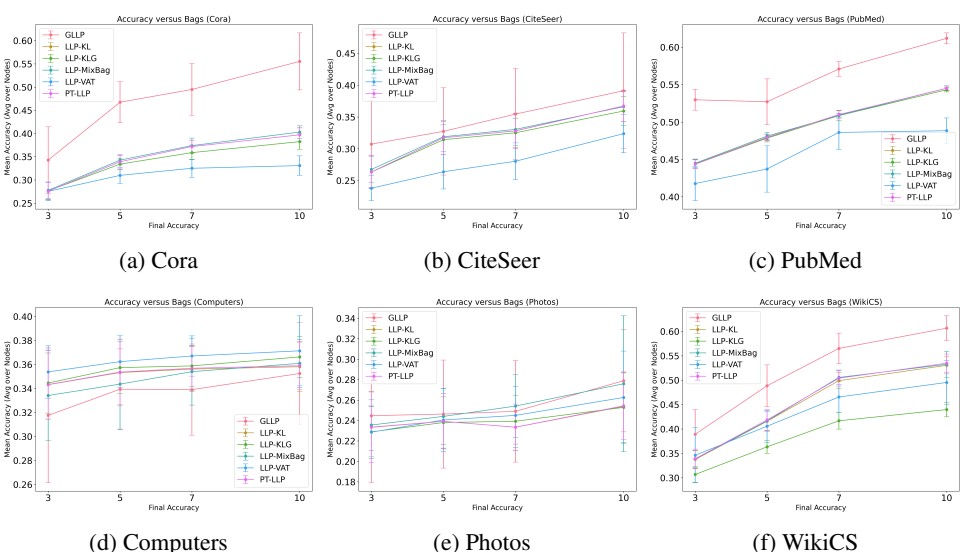

(a) Cora

(b) CiteSeer

(c) PubMed

(d) Computers

(e) Photos

(f) WikiCS

Figure 8: Test Accuracy versus number of bags for six datasets with graphsage backbone. The $x$-axis shows the number of bags labeled and $y$-axis shows the average final test accuracy over five seeds. Each method is plotted with mean and standard deviation bar.

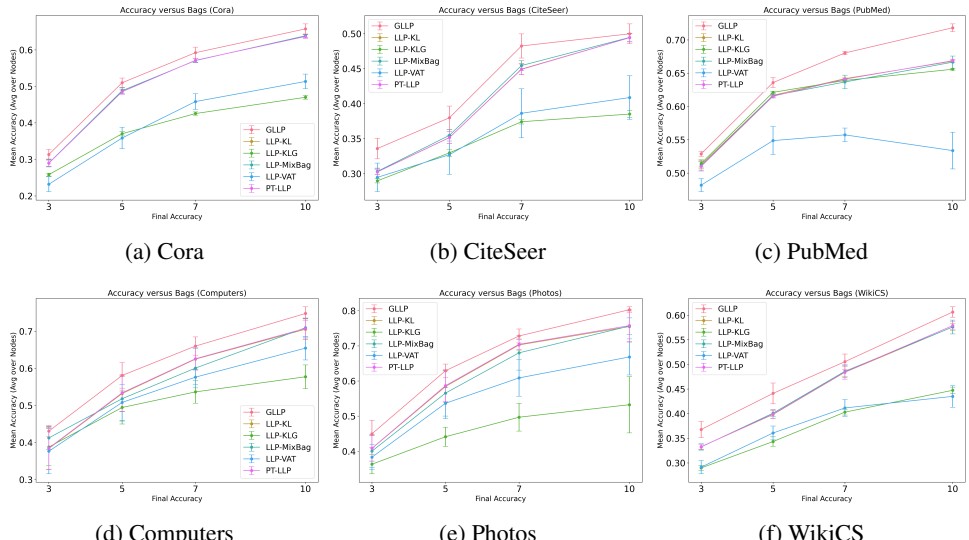

Figure 9: Test Accuracy versus number of bags for six datasets using ego-khop sampling methods. The $x$-axis shows the number of bags labeled and $y$-axis shows the average final test accuracy over five seeds. Each method is plotted with mean and standard deviation bar.

Table 3: Optimal Temperature Regularization ($\tau$) for each dataset and bag size.

| Dataset | Bag 3 | Bag 5 | Bag 7 | Bag 10 |
|---------|-------|-------|-------|--------|
| Cora | 2 | 2 | 2 | 2 |
| CiteSeer | 2 | 2 | 2 | 2 |
| PubMed | 2 | 2 | 2 | 2 |
| Computers | 4 | 2 | 2 | 2 |
| Photos | 2 | 1 | 2 | 2 |
| WikiCS | 2 | 2 | 2 | 2 |

## A.3 ADDITIONAL RESULTS

In Figure 6, we provide our results in terms of the number of nodes per bag with the average results across over different number of bags and seeds. It can be observed that our method GLLP still outperforms other baselines by a large margin, suggesting the robustness of our method under different settings. In Figure 7, we show the hyperparameter $\lambda$ and $\beta$ shows various patterns in terms of the number of nodes per bags, suggesting that the number of nodes per bag is not a proper supervision indicator reflecting model performance. In Figure 8, we provide results that leverages graphsage as backbone model for GNN training and we show that similar to results trained over GCN, our method still outperforms other baselines by large margin over most datasets. Figure 9 shows our results over bag generated through ego-khop sampling rather than random sampling, we show that our method is superior to other methods, suggesting its strong adaption to different sampling strategy over different graph datasets. Table 3 shows our hyperparameter $\tau$'s effect over different datasets and bag sizes and we show that $\tau = 2$ provides the best results overall and therefore we fixed it to 2 in the experiments.

### A.3.1 CASE STUDY: COVID OUTBREAK

To verify the effectiveness of our method in terms of real-world application, we conduct a case study on a simulated Covid outbreak dataset. Specifically, the dataset Covid generated from simulator of Covid breakout is provided by CovidSyn Wu & Nordling (2025), which is based on the real world covid breakout data Wu & Nordling (2024). The dataset provide a graph with nodes representing each person and edges representing their social contact during the pandenmic. It is known due to privacy reason, government or health center would not provide individual person's infection infor-

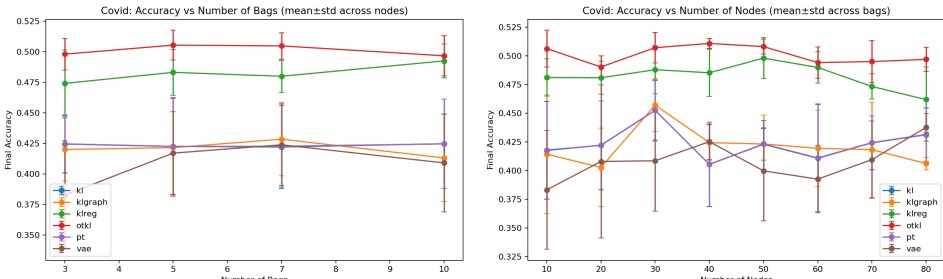

(a) Test accuracy versus number of bags per node (b) Test accuracy versus number of nodes per bag

Figure 10: Test Accuracy versus number of nodes per bag and number of bags per node for Covid Dataset. (a): the $x$-axis shows the number of bags used; (b): the $x$-axis shows the number of nodes used, and the $y$-axis shows the average final test acc over five seeds across different number of nodes for (a) and different number of bags for (b). Each method is plotted with mean and standard deviation bar.

mation (for example, infection state, health state) to the public but only the aggregated statistics such as the percentage of the infection and death rate. For the public who wants to know whether they are likely to be infected or not, it would be good to publish a blackbox model that allows individual to feed into their personal information and obtain a possible infection rate, which fits into our proposed problem scheme perfectly. Note that due to the edges formed by social contact, the graph is homophilic which also fits our theoretical assumption. We test our model against other baselines over the generated dataset. The results are shown in Figure 10 where our results consistently outperforms other baselines similar to other synthetic datasets.

