# OpenReview forum: "GLLP: Graph Learning from Label Proportions"
_ICLR.cc/2026/Conference — Submitted to ICLR 2026_

### Official Review · Reviewer_KUap · 2025-10-27

**Soundness:** 3
**Presentation:** 3
**Contribution:** 3
**Rating:** 4
**Confidence:** 2

**Summary:**

This paper studies the problem of learning from label proportion (LLP) for graph data. In particular, the paper devises a very interesting method GLLP, which decomposes the LLP problems into two: one is to explore optima transfer (OT) to derive soft pseudo-labels for nodes based on a bag-level label distribution and the node-level prediction from GNN models; the other is to train the GNN model based on the derived pseudo-labels. GLLP has alternations between these two steps, pseudo-label generation and GNN training, which is rather effective to iteratively refine node-level predictions and remain consistent with bag-level proportions. Experimental evaluation gives clear evidence to demonstrate the strength of the GLLP over other different proposals.

**Strengths:**

-- The paper addresses an important problem of LLP for graph data.
-- It presents a first proposal for the research problem and the main idea is quite appealing to me.
-- The paper is well-written and the experimental results give strong support for the proposed method.

**Weaknesses:**

-- The theoretical analysis is Section 3.3 is quite interesting to me, but the proof in the appendix is rather difficult for me to understand.
-- There are quite some typos in the paper: (page 3, line 125; page 4, line 194; page 5, line 228, line 236, line 248, line 254, line 260; page 6, line 299, etc. also a few typos on page 12).

**Questions:**

1. What is exactly the temperature \tau? In the experiments, it is simply set at 2. Why? How does its value impact the results?
2. In general, can you make the proof on page 12 more detailed?

Especially, how to expand Eq. 12 into Eqs. 26 and 27? I also don't understand the last sentence "Leveraging the assumption in Eq. 15, we can obtain the final conclusion". I thought Eq. 15 is what you aim to prove.

Once these are resolved, I am willing to improve my overall rating.

---

> ### Author Response · Authors · 2025-11-22
> **Response to Reviewer KUap 1/2**
>
> **1. Comment:  The theoretical analysis in Section 3.3 is quite interesting to me, but the proof in the appendix is rather difficult for me to understand. I also don't understand the last sentence "Leveraging the assumption in Eq. 15, we can obtain the final conclusion". I thought Eq. 15 is what you aim to prove.**
>
> ***Response:***
>
> Thank you for detailed comments on theoretical analysis. We wish to clarify that Eq. 15 is a consequence of homophilic assumption, but not our final conclusion. We leveraged this equation to obtain the final conclusion as Eq. 16.
>
> We use the standard edge homophilic assumption, where labels are more likely to be the same for nodes connected. Many studies have shown that under the homophilic graph data, averaging over neighbors prediction improves prediction accuracy when the final aggregation is not oversmoothed (i.e, finite steps of aggregation is beneficial, see for example [1] [2]).
>
> As a result, for homophilic graphs, we can argue that if we smooth our node's pseudo label based on its neighbors, the expected prediction accuracy should also be better than raw pseudo label, which is, in fact, what Eq. 15 is stating.
>
> **2. Comment: In general, can you make the proof on page 12 more detailed? Especially, how to expand Eq. 12 into Eqs. 26 and 27?**
>
> ***Response:***
>
> In the following, we provide a general sketch on how we obtain Eq.26 and Eq.27 from Eq. 12. In general, we have two steps to complete our proof.
>
>  ***First***, we highlight the main OT problem in the proof. Eq. 12 is essentially a minimization problem with constraints and our aim is to obtain the optimal `T^{*}` ($T^{opt}$) that solves the problem. After obtaining `T^{*}(\lambda)`, we denote $T^{opt}(\lambda)$ here, we normalize it to obtain the pseudo-label $\tilde{Y}(t)$ with t a change in variable for $\lambda$.  We note that Eq. 13 and its compact form Eq.14 (or Eq. 18 and Eq. 19 in appendix) is the direct solution of setting partial differential equation $\frac{\partial{L}}{\partial{T_{ij}}}=0$ under the Lagrangian:
>  $$
> \mathcal{L}(T,\boldsymbol{\alpha},\boldsymbol{\kappa}) = \sum_{i,j} C_{ij}^{\lambda} T_{ij} + \epsilon \sum_{i,j} T_{ij} \log T_{ij} + \sum_{i} \alpha_i \left( 1 - \sum_{j} T_{ij} \right) + \sum_{j} \kappa_j \left( n q_j - \sum_{i} T_{ij} \right)
> $$
> Recall the solution Eq.14 (Eq.19):
> $$
> T^{\text{opt}} (\lambda) = \text{Diag}(\boldsymbol{\alpha})\ \exp\left(-\frac{C^b}{\epsilon}\right)\ \exp\left(\frac{\lambda}{\epsilon} R Q^\tau\right)\ \text{Diag}(\boldsymbol{\kappa})
> $$
> Focus on a single row $T_i^{opt}(\lambda)$, which represents the soft pseudo-label (i.e., class distribution) for node $i$. Drop the diagonal scaling momentarily and isolate the exponential kernel:
> $$
> T_i^{\text{opt}}(\lambda) \propto
> \exp(-C^b_{i,:} \epsilon) \circ
> \exp((\lambda \epsilon)(R Q^\tau_{i,:}))
> $$
> At $\lambda = 0$, the solution becomes:
> $$
> T_i^{opt} (0) \propto \exp\left(-\frac{C^b_{i,:}}{\epsilon}\right)
> $$
> Therefore:
> $$
> T_i^{opt} (\lambda) \propto T_i^{opt} (0) \circ \exp\left(\frac{\lambda}{\epsilon} (RQ^\tau_{i,:})\right)
> $$
> This is a multiplicative reweighting of the baseline transport $T_i^{opt} (0)$ by a graph-aware factor.
> Now, we make a change of variable using t for $\frac{\lambda}{\epsilon}$
> $$
> t := \frac{\lambda}{\epsilon \gamma} \quad\text{with}\quad \gamma \ge \frac{\lambda}{\epsilon} \Rightarrow t \in [0,1]
> $$
> Now:
> $$
> \frac{\lambda}{\epsilon} = t\gamma \quad \Rightarrow \quad \exp\left(\frac{\lambda}{\epsilon} RQ^\tau\right) = \exp\left(t \cdot \gamma RQ^\tau\right)
> $$
> Hence:
> $$
> T_i^{opt} (\lambda) \propto T_i^{opt} (0) \circ \exp \left(t \cdot \gamma (RQ^\tau_{i,:})\right)
> $$
> To make $T_i^{opt} (\lambda)$ a valid soft label distribution, we need to apply softmax normalization:
> $$
> \tilde{Y} (t) = \text{Softmax} \left(\log T^{opt} (0) + t \cdot \gamma (RQ^\tau) \right)
> $$
> Let the second probability distribution (which captures the graph influence) be:
> $$
> r := \text{Softmax}(\gamma (RQ^\tau_{i,:}))
> $$
> We know that $log(r)$ is equivalent to $\gamma (RQ^\tau_{i,:})$ up to a constant offset, which can be canceled when put into the softmax function. To ensure that when t=1, the softmax produce the desired endpoint distribution r, we rescale $\tilde{Y} (t)$ to:
> $$
> \tilde{Y} (t) := \text{Softmax} \left((1-t)\log T_{i}^{opt}(0) + t \cdot log(r) \right)
> $$
> Then the final soft label distribution $\tilde{Y} (t)$ can be expressed as an Exponential Convex Combination of the baseline transport $T_i^{opt}(0)$ and the graph probability distribution $r$:
> $$
> \tilde{Y} (t) = \frac{T_{ik}^{opt} (0)^{1 - t} \cdot r_k^t}{\sum_j T_{ij}^{opt} (0)^{1 - t} \cdot r_j^t}
> $$
> which is Eq. 24.

---

> ### Author Response · Authors · 2025-11-22
> **Response to Reviewer KUap 2/2**
>
> **2. Comment: In general, can you make the proof on page 12 more detailed? Especially, how to expand Eq. 12 into Eqs. 26 and 27? (continued)**
>
> ***Response (continued):***
>
> ***Second***, we evaluate the quality of our generated pseudo-label in terms of $t>0$ versus $t=0$ by setting the metric as KL divergence of ground truth label and the generated pseudo-label, which we then leveraged our assumption defined in Eq. 15. Our metric $KL(a||b)$ is stated in Eq. 25 in the original paper appendix, and Eq. 26 is the direct consequence of substitution of $a=Q_i^*$ and $b=\tilde{Y}_{ik}(t)$.
>
> ***Finally***, we leverage the convex inequality from KL and the assumption Eq. 15 to obtain the final inequality.
> Because markdown does not support certain notation render, please refer to our revised submission for a more detailed description.
>
> **3. Comment: What is exactly the temperature $\tau$? In the experiments, it is simply set at 2. Why? How does its value impact the results?**
>
>  ***Response:***
>
> *The temperature* $\tau$ *controls how confident the prediction would be for each node. A higher* $\tau$ *would smooth the probability for each class and represent a less confident prediction for a specific class.*
>
> *In general, without strong supervision, we expect the model output to be slightly weak, and therefore we use a slightly higher* $\tau$ *to reduce the possibility of strong error signals. We provide a table that outputs the averaged best* $\tau$ *for 5 seeds for Cora, Citeseer, and PubMed dataset and we observe that* $\tau=2$ *is the optimal choice for these datasets. We will provide all the dataset results in the revision.*
>
> ---
> ### Table 10: shows the averaged accuracy and standard deviation achieved by the best performing hyperparameter setting for each $\tau$ value, averaged across all seeds, for three dataset CiteSeer, Cora, and PubMed.
> |   $\tau$ | CiteSeer   | Cora  | PubMed  |
> |--------------:|:------------------------|:--------------------|:----------------------|
> |             1 | 0.432 ± 0.074         | 0.456 ± 0.045     | 0.536 ± 0.056       |
> |             2 | **0.553 ± 0.071**         | **0.503 ± 0.060**     | **0.577 ± 0.011**       |
> |             4 | 0.367 ± 0.027         | 0.322 ± 0.008     | 0.536 ± 0.026       |
> |             8 | 0.313 ± 0.008         | 0.313 ± 0.003     | 0.519 ± 0.021       |
> |            10 | 0.311 ± 0.006         | 0.311 ± 0.003     | 0.517 ± 0.020       |
> ---
>
> **4. Comment: There are quite some typos in the paper: (page 3, line 125; page 4, line 194; page 5, line 228, line 236, line 248, line 254, line 260; page 6, line 299, etc. also a few typos on page 12).**
>
> ***Response:***
>
> Thank you for pointing out the typos, which have been fixed in the revision. We also carefully proofread the paper and fixed similar typos.
>
> **Reference**
> [1] Nicolas Keriven (2022). Not too little, not too much: a theoretical analysis of graph (over)smoothing. In The First Learning on Graphs Conference.
> [2] Daniel Winter, Niv Cohen, & Yedid Hoshen. (2024). Classifying Nodes in Graphs without GNNs.  arXiv:2402.05934

---

> > ### Comment · Reviewer_KUap · 2025-11-26
> >
> > Thank you for your clarifications.

---

### Official Review · Reviewer_PRqa · 2025-10-29

**Soundness:** 2
**Presentation:** 3
**Contribution:** 2
**Rating:** 2
**Confidence:** 4

**Summary:**

This paper introduces Graph Learning from Label Proportions (GLLP), extending the Learning from Label Proportions (LLP) paradigm to graph-structured data for node classification tasks. The authors highlight that the bag-level supervision yields weak signals. To address this, they propose to leverage optimal transport and graph penalty terms, with soft pseudo-labels.

**Strengths:**

1. The paper study an underexplored LLP problem in graph domain.
2. The authors propose to generate pseudo-label based on graph structure inductive bias.
3. The writing is easy to understand.

**Weaknesses:**

* While the introduction emphasizes LLP's applicability in scenarios where node-level labels are infeasible or undesirable, the experiments rely on standard homophilic benchmarks like citation networks (e.g., Cora, CiteSeer), Amazon product graphs, and WikiCS, which do not inherently reflect these constraints. Labels are readily available in these datasets, potentially undermining the method's real-world validation.
* The proposed method generates pseudo-labels based solely on bag-level proportions and graph structure. Intuitively, for a given bag and its proportion, there is a vast solution space of possible node label assignments that are consistent with both the proportion and the graph's homophily. The paper would be strengthened by a deeper investigation into this ambiguity, such as an analysis of the stability of the pseudo-labels or the sensitivity of the final results to different initializations, which is currently lacking.
* Figure 3, intended to illustrate the Optimal Transport process, is not sufficiently detailed to enhance reader understanding.
* For Theorem 1, the critical homophily assumption is not formally defined in the main text. The statement "Under the homophily assumption..." is vague.
* The experimental setup follows prior LLP works by using random sampling to create bags. However, in a graph context, random sampling can be detrimental as it may arbitrarily fracture local community structures.
* Only evaluate a single GCN backbone.
* The code is not available.

**Questions:**

* More evaluation on data with consistent and reasonable constraints.
* Detailed theorem presentation.
* Reproducible issues.

**Details Of Ethics Concerns:**

N/A.

---

> ### Author Response · Authors · 2025-11-22
> **Response to Reviewer PRqa 1/2**
>
> **1. Comment: While the introduction emphasizes LLP's applicability in scenarios where node-level labels are infeasible or undesirable, the experiments rely on standard homophilic benchmarks like citation networks (e.g., Cora, CiteSeer), Amazon product graphs, and WikiCS, which do not inherently reflect these constraints. Labels are readily available in these datasets, potentially undermining the method's real-world validation.**
>
>    ***Response:***
>
> Thank you for raising concern about our limited datasets usage. As pointed out by* **Reviewer Fxuc** *and* **Reviewer sQW4**, *we have gathered a new dataset concerning Covid prediction to showcase real-world application of graph LLP learning, which fits our problem definition and assumptions. Please see Response for* **Reviewer sQW4** *Comment 1* regarding Covidsyn dataset, graph LLP learning task, and result comparisons.
>
> **2. Comment: The proposed method generates pseudo-labels based solely on bag-level proportions and graph structure. Intuitively, for a given bag and its proportion, there is a vast solution space of possible node label assignments that are consistent with both the proportion and the graph's homophily. The paper would be strengthened by a deeper investigation into this ambiguity, such as an analysis of the stability of the pseudo-labels or the sensitivity of the final results to different initializations, which is currently lacking.**
>
>    ***Response:***
>
> Thank you for raising concerns about the stability of the generated pseudo-label. We want to clarify that the quality of the generated pseudo-label dependents on the entire pipeline as they are dynamically updated and GNN's output could affect its quality. Therefore, we can compare its sensitivity in terms of our hyperparameter analysis.
>
> In the paper Figure 5 (a) and (b), we have shown that for different settings (various number of bags), the best hyperparameter averaged over 5 seeds varies. So the quality of the pseudo-label depends on the choice of hyperparameters, as shown in Figure 5.
>
> **3. Comment: Figure 3, intended to illustrate the Optimal Transport process, is not sufficiently detailed to enhance reader understanding.**
>
>    ***Response:***
>
> In the revision, we have added step-wise description in the diagram (Figure 3) to better reflects each step of the OT process. Please refer to the revision for detailed descriptions.
>
> **4. Comment: For Theorem 1, the critical homophily assumption is not formally defined in the main text. The statement "Under the homophily assumption..." is vague.**
>
>    ***Response:***
>
> Thank you for your concern on our theoretical analysis. Please check* **Reviewer KUap** *comment 1's detailed response for our theoretical analysis clarification.
>
> **5. Comment: The experimental setup follows prior LLP works by using random sampling to create bags. However, in a graph context, random sampling can be detrimental as it may arbitrarily fracture local community structures.**
>
>    ***Response:***
>
> Thank you for raising concern about the sampling strategy choice in our work. To address the reviewer's concern, we have added additional sampling strategy, k-hop, for bag generation. In this case, each bag is not generated by using random sampling, but using nodes from k-hop neighbors. We have reported results in the response to*  **Reviewer sQW4** *Comment 2 (the Tables 4-6 show detailed performance and comparisons).

---

> ### Author Response · Authors · 2025-11-22
> **Response to Reviewer PRqa 2/2**
>
> **6. Comment: Only evaluate a single GCN backbone.**
>
>    ***Response:***
>
> Thank you for raising the concern about the backbone choice for our framework. Our empirical study suggests that different backbones show little effect in terms of model performance due to the lack of strong supervision signals, which makes us to decide a simple and parameter-saved graph convolution network model. Similar to unsupervised learning tasks, our empirical finding show that most of performance gain come from manipulating or processing the weak signal to obtain a good-quality strong signals rather than changing the base model directly.
>
> To address the reviewer's concerns, we report results using graphsage as backbone for Cora dataset under different settings in Tables 7-9. The results show similar trend with our GLLP method, demonstrating consistent and significant performance gain. We will include all dataset results in the Appendix in the camera-ready version.
>
> ---
> ### Table 7: Results of using graphsage as backbone for the Cora dataset under different bag sizes.
> | Method / # of bags | 3 | 5 | 7 | 10 |
> | --- | --- | --- | --- | --- |
> | LLP-KL | 0.28 ± 0.06 | 0.34 ± 0.03 | 0.37 ± 0.02 | 0.40 ± 0.03 |
> | LLP-KLG | 0.28 ± 0.06 | 0.33 ± 0.03 | 0.36 ± 0.02 | 0.38 ± 0.03 |
> | LLP-MixBag | 0.28 ± 0.06 | 0.35 ± 0.04 | 0.38 ± 0.02 | 0.41 ± 0.03 |
> | PT-LLP | 0.28 ± 0.06 | 0.34 ± 0.03 | 0.37 ± 0.02 | 0.40 ± 0.03 |
> | LLP-VAT | 0.28 ± 0.05 | 0.31 ± 0.02 | 0.33 ± 0.01 | 0.33 ± 0.03 |
> | GLLP | **0.37 ± 0.10** | **0.48 ± 0.14** | **0.50 ± 0.10** | **0.57 ± 0.10** |
> ------------------------------------------------------------------------
> ### Table 8: Results of using graphsage as backbone for the Cora dataset under different node sizes.
> | Method / # of nodes per bag | 10 | 20 | 30 | 40 | 50 | 60 | 70 | 80 |
> | --- | --- | --- | --- | --- | --- | --- | --- | --- |
> | LLP-KL | 0.30 ± 0.09 | 0.30 ± 0.07 | 0.36 ± 0.06 | 0.35 ± 0.05 | 0.37 ± 0.05 | 0.38 ± 0.05 | 0.34 ± 0.02 | 0.37 ± 0.03 |
> | LLP-KLG | 0.30 ± 0.10 | 0.30 ± 0.07 | 0.35 ± 0.06 | 0.35 ± 0.04 | 0.36 ± 0.05 | 0.37 ± 0.04 | 0.33 ± 0.01 | 0.34 ± 0.01 |
> | LLP-MixBag | 0.31 ± 0.10 | 0.31 ± 0.07 | 0.36 ± 0.06 | 0.36 ± 0.05 | 0.38 ± 0.06 | 0.39 ± 0.05 | 0.34 ± 0.03 | 0.38 ± 0.03 |
> | PT-LLP | 0.30 ± 0.10 | 0.30 ± 0.07 | 0.36 ± 0.06 | 0.35 ± 0.05 | 0.37 ± 0.05 | 0.38 ± 0.05 | 0.34 ± 0.02 | 0.37 ± 0.03 |
> | LLP-VAT | 0.30 ± 0.07 | 0.30 ± 0.07 | 0.32 ± 0.03 | 0.31 ± 0.01 | 0.33 ± 0.02 | 0.32 ± 0.01 | 0.30 ± 0.00 | 0.31 ± 0.00 |
> | GLLP | **0.38 ± 0.05** | **0.30 ± 0.10** | **0.45 ± 0.09** | **0.46 ± 0.09** | **0.53 ± 0.11** | **0.54 ± 0.12** | **0.53 ± 0.08** | **0.64 ± 0.07** |
> ---------------------------------------------------------------------
> ### Table 9: Results of using graphsage as backbone for the Cora dataset
> | Method / Dataset | Cora |
> | --- | --- |
> | LLP-KL | 0.35 ± 0.06 |
> | LLP-KLG | 0.34 ± 0.05 |
> | LLP-MixBag | 0.35 ± 0.06 |
> | PT-LLP | 0.35 ± 0.06 |
> | LLP-VAT | 0.31 ± 0.04 |
> | GLLP | **0.48 ± 0.13** |
> ---
>
> **7. Comment: The code is not available.**
>
>    ***Response:***
>
> We are fully committed to publish code and data for public access. For reviewers' validation, we have provided the code along with the newly generated dataset all through the following anonymous Gitub: https://anonymous.4open.science/r/GLLP-2C9E/README.md.

---

### Official Review · Reviewer_FxuC · 2025-10-31

**Soundness:** 3
**Presentation:** 3
**Contribution:** 3
**Rating:** 6
**Confidence:** 4

**Summary:**

This paper studied a new problem, i.e. LLP, in graph domain, where we only know the label distributions over each bag of nodes, rather than node-level labels. The authors proposed two-level losses, the first is bag-level KL loss, and the second is node-level supervison, got by OT. Experiments showed good performances of the technique.

**Strengths:**

1. The authors involved an interesting problem, Learning from Lable Proportion, into graph domain.
2. The theoretical and experimental stuff supports their techniques.
3. The organization is clear and easy to follow.

**Weaknesses:**

1. Learning from Label proportion sounds more like an industrial scenario. For users' privacy, we have to mask sensitive labels. Also in introduction, the authors mentioned online advertising. But all experments were done on non-industrial datasets. The used datasets, like Cora, Citeseer, never have the requirements of masking labels. So, I suggest the authors to try their techniques on some industrial cases.

2. The authors used optimal transport to get pssudo-labels. OP has a high cost O(N^2). I concern about scalability and efficiency if the authors compute OT every epoch.

**Questions:**

As in Eq. (9), the cost matrix C is obtained based on logits, i.e., the GNN output. But if GNN is not well-trained, especially at the early stage, the quality of the cost matrix could be low. Why not try to decompose bag-level label proportions into node-level pesudo-signals, and use these signals, rather than logits, to construct cost matrix?

---

> ### Author Response · Authors · 2025-11-22
> **Response to Reviewer Fxuc**
>
> **1. Comment: Learning from Label proportion sounds more like an industrial scenario. For users' privacy, we have to mask sensitive labels. Also in introduction, the authors mentioned online advertising. But all experments were done on non-industrial datasets. The used datasets, like Cora, Citeseer, never have the requirements of masking labels. So, I suggest the authors to try their techniques on some industrial cases.**
>
>    ***Response:***
>
> Thank you for your advice on testing over industrial datasets. We have conducted experiments over Covidsyn dataset to showcase real-world applications of graph LLP learning. Please refer to the Response to*  **Reviewer sQW4** *Comment 1* regarding Covidsyn dataset, graph LLP learning task, and result comparisons.
>
> **2. Comment: The authors used optimal transport to get pseudo-labels. OP has a high cost $O(N^2)$. I concern about scalability and efficiency if the authors compute OT every epoch.**
>
>    ***Response:***
>
> Thank you for raising concern about the computation cost on OP computation. We never used N in the paper, so we assume that the reviewer refers to N as the lower case n we used in the problem definition, where n denotes the number of nodes in the graph.
>
> With respect, we wish to clarify that our OP cost is not* $O(N^2)$ *but* $O(kb_{max}^2)$ *where b_{max} denotes the maximum number of nodes within bags and k is the total number of bags. As * $b_{max} << N$ *and in our problem setting we consider supervision to be extremely rare with the number of bags k ranging from [3,10] and the number of nodes within the bag n ranging from [20,80]. As a result, the computation cost is negligible compared to GNN's computation cost. Additionally, we use the Sinkhorn OT with its entropic version with typically only 30-50 iterations (empirically tested) to converge and is much faster compared with traditional solver (theoretically proved by [1]).
>
> **3. Comment: As in Eq. (9), the cost matrix C is obtained based on logits, i.e., the GNN output. But if GNN is not well-trained, especially at the early stage, the quality of the cost matrix could be low. Why not try to decompose bag-level label proportions into node-level pseudo-signals, and use these signals, rather than logits, to construct cost matrix?.**
>
>  ***Response:***
>
> Thank you for raising concern about the low quality GNN output in the initial stage. We want to point out that the bag-level label proportion signal itself is noisy and weak when applied to node-level pseudo-signals.
>
> A simple idea to realize such condition could be to fully train a GNN model over the bag-level label proportion signal such by using KL divergence loss and then produce a node-level pseudo-signal for further OT training.
>
> In our paper, PT-LLP adopts a similar idea, and the results show that this approach leads to poor performance, suggesting that it is difficult to first train a GNN model that outputs relatively good node-level signals using only bag-level signals. Alternatively, the proposed GLLP employs an iterative style update that occurs for each epoch.
>
> **Reference**
> [1] M. Cuturi, “Sinkhorn distances: Lightspeed computation of optimal transport,” in Proc. Adv. Neural Inf. Process. Syst. (NIPS), 2013, pp. 2292–2300.

---

### Official Review · Reviewer_sQW4 · 2025-11-04

**Soundness:** 2
**Presentation:** 3
**Contribution:** 2
**Rating:** 2
**Confidence:** 4

**Summary:**

This paper introduces GLLP, a framework that extends Learning from Label Proportions (LLP) to graph-structured data, where nodes are interdependent. The method employs Optimal Transport with a homophily-aware cost to generate soft pseudo-labels for nodes, enabling effective node classification under distributional supervision. Theoretical analysis and experiments demonstrate that GLLP outperforms existing LLP baselines.

**Strengths:**

1. Applying LLP to graph structures is a novel contribution.

2. The proposed method is consistent with the theoretical analysis.

**Weaknesses:**

1. My main concern is the lack of real-world scenarios for the proposed graph LLP problem. The paper does not present convincing real-world applications, and the experiments are conducted only on synthetic datasets.

2. For the proposed graph LLP setting, the paper should discuss more carefully the role of edges — both between different bags and within each bag. Without such analysis, simply applying the LLP framework to graph data is not particularly meaningful.

3. Using only synthetic datasets in the experiments is insufficient to validate the practicality and relevance of the proposed graph LLP setting.

**Questions:**

on weakness

---

> ### Author Response · Authors · 2025-11-22
> **Response to Reviewer sQW4 1/2**
>
> **1. Comment: My main concern is the lack of real-world scenarios for the proposed graph LLP problem. The paper does not present convincing real-world applications, and the experiments are conducted only on synthetic datasets. (Using only synthetic datasets in the experiments is insufficient to validate the practicality and relevance of the pred graph LLP setting.)**
>
>    ***Response***
>
> We thank the reviewer for commenting on the real-world graph LLP application. Following the suggestions, we have identified a new Covid prediction dataset, CovidSyn [1], which is based on Covid outbreak prediction [2], as a graph LLP learning task.
>
> The CovidSyn dataset is represented as a graph with nodes representing each person and edges representing their social contact during the pandemic. Due to privacy reasons, government or health centers cannot provide individual person's infection information (such as infection or health status) to the public, but can release aggregated statistics such as the percentage of infection or death rate within a region. In this case, the Covid infection information is published as aggregated label proportions. For individuals or local health administration who want to estimate the risk of being infected or not, the graph LLP learning allows them to leverage personal information and label proportions to derive a possible infection rate (i.e., a graph LLP learning task). Note that, due to the edges formed by social contact, the graph is homophilic, which also fits our theoretical assumption.
>
> In the revision, we have tested our model against other baselines over the CovidSyn dataset and reported the results in the paper's Appendix A.3.1 "Case Study: Covid Outbreak".
>
> Tables 1-3 below also briefly report the model performance on the CovidSyn dataset.
>
> ---
> ### Table 1: Results of using GCN for CovidSyn dataset under different bag sizes.
>   | Covid | 3 | 5 | 7 | 10 |
> | --- | --- | --- | --- | --- |
> | LLP-KL | 0.42 ± 0.02 | 0.42 ± 0.04 | 0.42 ± 0.03 | 0.42 ± 0.04 |
> | LLP-KLG | 0.42 ± 0.03 | 0.42 ± 0.04 | 0.43 ± 0.03 | 0.41 ± 0.04 |
> | LLP-MixBag | 0.47 ± 0.03 | 0.48 ± 0.02 | 0.48 ± 0.01 | 0.49 ± 0.01 |
> | PT-LLP | 0.42 ± 0.02 | 0.42 ± 0.04 | 0.42 ± 0.03 | 0.42 ± 0.04 |
> | LLP-VAT | 0.38 ± 0.04 | 0.42 ± 0.03 | 0.42 ± 0.03 | 0.41 ± 0.04 |
> | GLLP | **0.50 ± 0.01** | **0.51 ± 0.01** | **0.50 ± 0.01** | **0.50 ± 0.02** |
> ----------
> ### Table 2: Results of using GCN for CovidSyn dataset under different node sizes.
> | Covid | 10 | 20 | 30 | 40 | 50 | 60 | 70 | 80 |
> | --- | --- | --- | --- | --- | --- | --- | --- | --- |
> | LLP-KL | 0.42 ± 0.04 | 0.42 ± 0.04 | 0.45 ± 0.03 | 0.41 ± 0.04 | 0.42 ± 0.01 | 0.41 ± 0.05 | 0.42 ± 0.02 | 0.43 ± 0.02 |
> | LLP-KLG | 0.41 ± 0.05 | 0.40 ± 0.03 | 0.46 ± 0.02 | 0.42 ± 0.02 | 0.42 ± 0.03 | 0.42 ± 0.03 | 0.42 ± 0.04 | 0.41 ± 0.01 |
> | LLP-MixBag | 0.48 ± 0.02 | 0.48 ± 0.01 | 0.49 ± 0.02 | 0.49 ± 0.02 | 0.50 ± 0.02 | 0.49 ± 0.01 | 0.47 ± 0.01 | 0.46 ± 0.03 |
> | PT-LLP | 0.42 ± 0.04 | 0.42 ± 0.04 | 0.45 ± 0.03 | 0.41 ± 0.04 | 0.42 ± 0.01 | 0.41 ± 0.05 | 0.42 ± 0.02 | 0.43 ± 0.02 |
> | LLP-VAT | 0.38 ± 0.05 | 0.41 ± 0.07 | 0.41 ± 0.04 | 0.43 ± 0.02 | 0.40 ± 0.04 | 0.39 ± 0.03 | 0.41 ± 0.03 | 0.44 ± 0.01 |
> | GLLP | **0.51 ± 0.02** | **0.49 ± 0.01** | **0.51 ± 0.01** | **0.51 ± 0.00** | **0.51 ± 0.01** | **0.49 ± 0.01** | **0.50 ± 0.02** | **0.50 ± 0.01** |
> ---
> ### Table 3: Results of using GCN for CovidSyn dataset under all variants.
> | Method | Covid |
> | --- | --- |
> | LLP-KL | 0.42 ± 0.03 |
> | LLP-KLG | 0.42 ± 0.03 |
> | LLP-MixBag | 0.48 ± 0.02 |
> | PT-LLP | 0.42 ± 0.03 |
> | LLP-VAT | 0.41 ± 0.04 |
> | GLLP | **0.50 ± 0.01** |
> ---

---

> ### Author Response · Authors · 2025-11-22
> **Response to Reviewer sQW4 2/2**
>
> **2. Comment: For the proposed graph setting, the paper should discuss more carefully the role of edges — both between different bags and within each bag. Without such analysis, simply applying the LLP framework to graph data is not particularly meaningful.**
>
>    ***Response:***
>
> Thank you for your advice on the analysis over inter-bag and intra-bag edge difference. We argue that in the GNN stage, such difference can be safely ignored, since GNN can be trained over a single graph directly if without training over batches. When applied over KL loss over bag, we already convert the edge information to the embedding over nodes. When applied to Optimal transport, graph regularization does take advantage of the intra-bag edge information. In this case, we note that the edge information is mainly dependent on the bag sampling strategy and the structure of the graph.
>
> To address the reviewer's concerns, in Tables 4 to 6, we reported additional bag sampling strategy that specifically target for k-hop neighbors. In this case, each bag would have more edges, compared to inter-bag edges. The results suggest a similar performance, validating the effectiveness of the proposed GLLP method in leveraging topology inside each bag for graph learning :
>
> ---
> ### Table 4: Results for the Cora dataset under different bag sizes using k-hop neighbors sampling strategy for bag sampling.
> | Method / # of bags | 3 | 5 | 7 | 10 |
> | --- | --- | --- | --- | --- |
> | LLP-KL | 0.29 ± 0.12 | 0.49 ± 0.10 | 0.57 ± 0.08 | 0.64 ± 0.09 |
> | LLP-KLG | 0.26 ± 0.11 | 0.37 ± 0.05 | 0.43 ± 0.03 | 0.47 ± 0.03 |
> | LLP-MixBag | 0.29 ± 0.12 | 0.49 ± 0.10 | 0.57 ± 0.08 | 0.64 ± 0.09 |
> | PT-LLP | 0.29 ± 0.12 | 0.49 ± 0.10 | 0.57 ± 0.08 | 0.64 ± 0.09 |
> | LLP-VAT | 0.23 ± 0.10 | 0.36 ± 0.09 | 0.46 ± 0.05 | 0.51 ± 0.07 |
> | GLLP | **0.32 ± 0.11** | **0.51 ± 0.11** | **0.60 ± 0.10** | **0.66 ± 0.09** |
> ---
> ### Table 5: Results for the Cora dataset under different node sizes using k-hop neighbors sampling strategy for bag sampling.
> | Method / # of nodes per bag | 10 | 20 | 30 | 40 | 50 | 60 | 70 | 80 |
> | --- | --- | --- | --- | --- | --- | --- | --- | --- |
> | LLP-KL | 0.36 ± 0.10 | 0.48 ± 0.22 | 0.48 ± 0.06 | 0.53 ± 0.10 | 0.54 ± 0.22 | 0.60 ± 0.11 | 0.50 ± 0.23 | 0.49 ± 0.21 |
> | LLP-KLG | 0.33 ± 0.09 | 0.35 ± 0.14 | 0.42 ± 0.03 | 0.42 ± 0.06 | 0.38 ± 0.12 | 0.42 ± 0.04 | 0.38 ± 0.16 | 0.34 ± 0.14 |
> | LLP-MixBag | 0.37 ± 0.09 | 0.48 ± 0.23 | 0.48 ± 0.05 | 0.53 ± 0.11 | 0.54 ± 0.22 | 0.60 ± 0.11 | 0.50 ± 0.23 | 0.49 ± 0.21 |
> | PT-LLP | 0.36 ± 0.10 | 0.48 ± 0.22 | 0.48 ± 0.06 | 0.53 ± 0.10 | 0.54 ± 0.22 | 0.60 ± 0.11 | 0.50 ± 0.23 | 0.49 ± 0.21 |
> | LLP-VAT | 0.36 ± 0.10 | 0.35 ± 0.14 | 0.39 ± 0.04 | 0.34 ± 0.18 | 0.39 ± 0.17 | 0.42 ± 0.04 | 0.42 ± 0.20 | 0.45 ± 0.19 |
> | GLLP | **0.36 ± 0.10** | **0.49 ± 0.21** | **0.50 ± 0.07** | **0.56 ± 0.11** | **0.58 ± 0.22** | **0.61 ± 0.11** | **0.52 ± 0.24** | **0.57 ± 0.19** |
> ---
> ### Table 6: Results for the Cora dataset under all variants using k-hop neighbors sampling strategy for bag sampling.
> | Method / Ddataset | Cora |
> | --- | --- |
> | LLP-KL | 0.50 ± 0.16 |
> | LLP-KLG | 0.38 ± 0.10 |
> | LLP-MixBag | 0.50 ± 0.16 |
> | LLP-VAT | 0.39 ± 0.13 |
> | GLLP | **0.52 ± 0.16** |
>
> **Reference**
> [1] Y. Wu, T. Nordling. "CovSyn: an agent-based model for synthesizing COVID-19 course of disease and contact tracing data," in medRxi, 2025.
>
> [2] Y. Wu, T. Nordling. "A structured course of disease dataset with contact tracing information in Taiwan for COVID-19 modelling," in  Scientific Data, vol. 11, pp. 821, 2024.

---

> > ### Comment · Reviewer_sQW4 · 2025-11-26
> >
> > thanks for rebuttal, but i also keep the concern

---

### Meta-Review · Area_Chair_vHSD · 2025-12-09

**Summary:**

This paper studies node classification under label-proportion supervision on graphs and proposes an iterative framework that combines Optimal Transport–based pseudo-labeling with homophily-aware graph modeling. The idea is novel for graph-domain LLP and the method is clearly presented, with theoretical motivation and a reasonably broad set of experiments. Reviewers initially raised concerns about the lack of convincing real-world scenarios, limited analysis of the role of edges and bag construction, theoretical clarity, and restricted experimental settings. The authors responded with a new CovidSyn case study, additional sampling strategies and backbone experiments, clearer theoretical explanations, and released code; these revisions addressed several technical and clarity issues, although concerns about the practical relevance of the LLP setting and the stability or ambiguity of pseudo-labels remain only partially resolved.

**Reviewer Concerns:**

In the initial reviews, several concerns were raised. Reviewer KUap’s questions regarding the theoretical derivation, the role of the temperature parameter, and clarity issues were largely resolved through detailed rebuttal explanations. Reviewer FxuC’s concerns about OT complexity and backbone dependence were also addressed with complexity analysis and additional GraphSAGE results. Reviewer PRqa’s comments on sampling strategy, backbone diversity, and figure clarity were partially addressed through new k-hop sampling experiments, expanded backbone evaluations, and revised illustrations. Concerns from Reviewer sQW4 and PRqa about real-world applicability, the strength of the motivating scenario, and the ambiguity or stability of pseudo-label solutions remain outstanding, as the CovidSyn case study only partially alleviates the need for a compelling real-world dataset and the paper still lacks a deeper analysis of pseudo-label identifiability.

**Reviewer Scores:**

Reviewer sQW4 would likely keep the same score, as they stated that their main concerns remained. Reviewer FxuC would probably keep their moderately positive score, since their technical questions were satisfactorily addressed. Reviewer PRqa might increase their score slightly, though their core concerns about real-world relevance persist. Reviewer KUap would likely raise their score, given their stated willingness to do so after receiving the requested clarifications.

---

### Decision · Program_Chairs · 2026-01-26

Reject